# Families, Parenting and Aggressive Preschoolers: A Scoping Review of Studies Examining Family Variables Related to Preschool Aggression

**DOI:** 10.3390/ijerph192315556

**Published:** 2022-11-23

**Authors:** Raúl Navarro, Elisa Larrañaga, Santiago Yubero, Beatriz Víllora

**Affiliations:** Department of Psychology, Faculty of Education and Humanities, University of Castilla-La Mancha, Avda de los Alfares, 42, 16071 Cuenca, Spain

**Keywords:** preschool aggression, childhood, aggression, family, parenting, bullying, scoping review

## Abstract

Background: A growing body of research has shown that children behave aggressively from an early age. In recent decades, such behaviour has become a focus of scientific interest, not only because of the adverse consequences of these interactions, but also because high levels of aggression, especially at an early age, may be a risk factor for the use of other forms of aggression, such as bullying, later on during their development. These behaviours are related not only to individual characteristics, but also to peer relationships, teacher behaviours, school variables, family factors and cultural influences. Method: In order to find out which family variables have been researched in relation to preschool aggression and which family variables are associated with perpetration and victimisation, a scoping review was conducted in accordance with the PRISMA guidelines. Four databases (Web of Science, Scopus, PubMed and PsycINFO) were used to map the studies published between 2000 and 2022. Results: This scoping review included 39 peer-reviewed articles from an initial sample of 2002 of them. The majority of studies looked only at perpetration behaviours. The main family variables covered in the articles concern parental behaviours, adverse childhood experiences in the family environment, and the household structural and sociodemographic characteristics. Conclusion: This scoping review shows that different factors within the family environment increase the risk of developing aggressive and victimising behaviours in the preschool setting. However, the relationship between the family variables and preschool aggression is complex, and it may be mediated by other factors such as gender, child–teacher closeness or parent–child dyads.

## 1. Introduction

Aggression has been defined as “any form of behaviour that is intended to injure someone physically or psychologically” [1]. By this definition, there are two essential components to aggressive behaviour: the intent to harm another person or persons, and this intent takes the form of harmful behaviour. Aggressive behaviour is generally classified as physical (hitting, pushing and shoving, spitting, etc.), verbal (shouting, making threats, insulting, etc.,) and relational (spreading rumours, ostracism, exclusion, etc.) [2,3]. The scientific literature considers physical and verbal aggression as a form of overt or direct aggression [4], and relational aggression is considered as a form of indirect and social aggression [5].

A significant number of studies show that boys and girls behave aggressively towards others from a very early age [6,7]. More precisely, physically aggressive behaviour can be observed in young children in early childhood, and it seems to intensify by the ages of two and three [8,9]. However, childhood physical aggression tends to decline with age and assume ever more subtle forms that are less obvious to adults [10,11]. In fact, the research suggests that children around 30 months old exhibit relational aggression in their interactions with other children [12], which is easily distinguishable from physical aggression [13].

It is unclear whether children intend to cause physical or psychological harm in infancy, although some studies indicate that children as young as three–six years old understand what it means to harm others, and so their actions should be considered to be a form of aggressive behaviour [14]. In fact, some researchers also suggest that children may intentionally harm others physically and relationally during childhood, while also using similar means to achieve some goals without meaning to cause harm. Take the example of a child forcibly grabbing a toy they like from the hand of their playmate. In this case, the main goal is to get the toy, rather than to intentionally harm the playmate [15]. The existence of objectives other than causing harm does not diminish the aggressiveness of the behaviour. Instead, we find ourselves in the realm of proactive aggression, which is born out of self-interest and is aimed at achieving some end, as opposed to reactive aggression, which is rooted in anger and seeks to cause harm [16]. Along these lines, Roseth and Pellegrini [17] found that children are already using proactive aggression in childhood to intimidate others or achieve dominance within the peer group.

Proactive aggression involves greater premeditation and intentionality rather than reactive aggression, and it more closely resembles the characteristics of school bullying [18]. There is growing discussion in the scientific literature on whether the term bullying can be used when one is researching aggressive behaviour at the preschool stage [19]. Bullying is physically, verbally or psychologically aggressive behaviour that is deliberate and repeated over time. It is further characterised by a power imbalance between the perpetrator and the victim, where the victim is or feels in a weaker position due to differences in physical strength, social skills, social status and so on [20]. Various studies suggest that bullying behaviour can exist as early as the preschool stage [21,22]. In particular, the research shows that the typical features of bullying are already evident during childhood. These include observing repetitive behaviours, identifying different roles—victim, aggressor, bystander and defender [23]—and the fact that intentionality is already present in such displays of aggression [16]. However, other researchers have noted significant differences between preschool aggressive behaviour and the later stages of schooling or adolescence, such as the absence of gender differences among the victims, role instability [24,25] or difficulties in recognising peripheral roles such as the bystander or the defender [26,27].

Irrespective of the term that is used, the above data suggest that aggression is a major issue that needs to be addressed in the early years of schooling when children begin to interact with their peers and experiment with different social behaviours. In recent decades, such behaviour has become a focus of scientific interest. This is not only due to the associated adverse consequences, but also because high levels of aggression, especially at an early age, may be a risk factor for the use of other forms of aggression at the later stages of development, such as bullying itself [27,28,29].

The early developmental factors associated with these behaviours are the subject of mounting research, with many studies examining the cognitive and emotional factors underlying these early displays of aggressive behaviour [30,31,32,33,34]. The socioecological theory [35] suggests that children’s behaviour is influenced by four structures: the individual characteristics (biological and personal background) that have an impact on children’s behaviour and social relationships; the immediate social and physical environment (where primary socialisation takes place); community-related factors (the formal and informal social institutions and structures where relationships develop); the society at large (the economic and social environment, including cultural norms). Within the immediate social and physical environment, the family context is fundamental for children’s personal, social and educational development. This is the first socialisation context in which attitudes and behaviours that will guide social behaviour in other contexts such as school are learned and reinforced [36]. The past research has shown that aggressive behaviours are associated with different family variables, such as the parents’ attitudes toward violence, the parents’ moral disengagement, a lack of family communication, an absence of emotional support, a deficient level of family cohesion, low levels of parental acceptance of their children, high levels of coercion/imposition (rules, limits, punishments, etc.), and having problematic relationships with their siblings [14,18].

Insight into how family variables relate to aggressive behaviours during the pre-primary educational stages is crucial for gaining a better understanding of the associated risk factors and protective mechanisms that can improve the prevention and intervention methods [37,38] and prevent spillover to the later educational stages [39]. In light of this need, a scoping review is an ideal way to determine the nature and volume of research, and thus, gauge the current state of knowledge in this field. This scoping review therefore aims to analyse the peer-reviewed literature on aggressive behaviours among preschool children in school settings together with associated family factors, synthesise the available evidence and outline the future research priorities.

## 2. Materials and Methods

### 2.1. Design

We conducted a scoping review of the published scientific literature on aggressive behaviour in preschool settings and the family factors linked to such behaviour. Scoping reviews provide an overview of the research that has been conducted on a specific topic, describing the body of existing work, giving an insight into how the previous studies have been conducted, identifying the variables related to the topic that is under study and highlighting the gaps in the research [40]. In the past, scoping reviews have been used to assess the available evidence on the prevalence, measurement instruments, risk factors and consequences of different forms of aggression in children and adolescents. For example, Nelson et al. [41] used this approach to identify the validated instruments that measure aggression and bullying among pre-adolescents and children. Bender et al. [42] conducted a scoping review to find out the role that is played by guns in adolescent dating violence. Srinivasan et al. [43] employed this approach to map the research on bullying among children and adolescents in low- and middle-income countries. 

The design and development of this scoping review followed the five-step methodological framework outlined by Levac et al. [44]: (1) identifying the research question, (2) identifying the relevant studies, (3) selecting the studies, (4) the charting and data analysis stage and (5) summarising and reporting the results. The screening of the studies and the summarising and reporting of the articles finally included in the review are described in line with the Preferred Reporting Items for Systematic Reviews and Meta-Analyses extension for Scoping Reviews (PRISMA-ScR) [45]. The protocol for this review was not registered in advance.

#### 2.1.1. Stage 1: Research Question

This scoping review was guided by the following question: *what is known about the family variables related to peer aggression and victimisation in preschool settings?* Specifically, we wanted to find out about the forms of preschool aggression explored by the research, the family variables analysed in relation to preschool children’s aggression, the methodology used to analyse these associations and the family variables that are most related to perpetration and victimisation in the preschool context. Prior to the review process, aggression in preschool settings was defined as any act in which a child or group of children insults, hits, socially excludes or threatens other classmates in the school. Aggression therefore includes any act of a physical, verbal or relational nature. These acts can be both reactive and proactive, and the researchers may use various terms such as aggression, victimisation, peer aggression, unjustified aggression or bullying, among others. According to the Center for Disease and Control Prevention [46] *preschoolers* are children from three–five years old. However, the age range varies across countries, and they may be as young as two years old, and some children in their last year of preschool may have reached six years old before the end of the year, so for the purposes of this review, the children of preschool age included those aged 24–72 months. With regard to the family factors, the aim was to analyse the variables associated with parenting (parenting styles, support, attachment, communication, etc.), family structure (including siblings), socioeconomic level, parental educational attainment and other variables such as the existence of family disputes, interparental violence, child abuse and parental psychopathology. 

#### 2.1.2. Stage 2: Identification of Studies

We used four bibliographic databases for the review: Web of Science (WoS), Scopus (SP), PubMed (PM) and PsycINFO (PI). The literature search was conducted in July 2022, and it was restricted to English and Spanish language articles that were published during or after 2000. The search was restricted to three dimensions with a search string for each one (see Table 1 for all of the dimensions and terms used): participants (Dimension 1), preschool aggression (Dimension 2), family context (Dimension 3). To ensure that we conducted as complete a search as possible, variants or synonyms of the established search terms were used. The search was carried out by combining the title, abstract and keywords in the four databases, and it was performed in both English and Spanish. 

#### 2.1.3. Stage 3: Study Selection

The search was conducted by the first and last authors (RN and BV, respectively), who both have experience with the previous reviews [47]. They also screened and selected the articles resulting from each database search, discussing any questionable records until an agreement was reached. Any remaining doubts or disagreements were discussed and resolved with the other authors.

The studies were included in the final analysis if they met the following criteria (see Table 2): (1) The studies had to be published in English or Spanish, in peer-reviewed journals and be quantitative or quantitative in nature. (2) The study participants had to be aged between the ages of two (24 months) and six (72 months) years old, thus spanning the second cycle of preschool education. If the studies looking at several age groups were included, only the data relating to the relevant age range (two–six years old) were reviewed. When no age range was reported, two criteria were taken into account for inclusion in the study: firstly, that the authors explicitly stated that the study was conducted in a preschool (and not a primary school or above), and secondly, that the mean age of the participants was from two–six years old with a small standard deviation. In longitudinal studies, only the information linked to the established age range was reviewed. For example, if a study looked at children aged two–six years old, and it measured the same participants at the ages of nine and fourteen, only the baseline measurement was taken into account in analysing the relationship between the aggressive behaviour and the family variables. (3) The study participants had to be from the general population, and research involving specific groups or clinical samples was excluded. (4) The articles had to analyse aggressive peer interactions in the school environment. As previous studies have shown that the parents may not be reliable as the sole sources of information on aggression outside the family [48], those that only measured the parent-reported preschool aggression and did not include the teachers’ reports, the school observation procedures or the peer nomination methods were excluded. (5) The analyses in the study had to include one or more family variables as predictor variables. If the study contained other family variables, but they were included as control variables or covariates in the analyses, only the results of the family variables that were included as predictors were reported.

Figure 1 shows the PRISMA flow chart, illustrating the identification, screening and selection process. The search for the terms described above in all of the databases used yielded 2002 articles. A total of 281 duplicate results were eliminated, leaving 1721 for the first phase of screening via the title and abstract. In this first screening, 1523 articles were eliminated, leaving 198 articles for the full screening. A total of 159 articles were eliminated after fully screening them against the above research question and exclusion criteria. In the end, 39 articles met the inclusion criteria and were included in the review.

#### 2.1.4. Stage 4. Charting and Data Analysis

Garrad’s matrix method [49] was used to extract information from each of the studies that was included in the final selection and synthesise their findings. The matrix (see Table A1) includes the geographical location, the year of publication, the sample size and characteristics, the study objectives, the forms of preschool aggression analysed, the methodology used to measure preschool aggression, the family variables included in the analyses, the type of statistical analysis used to analyse the relationship between preschool aggression and family variables, and the main findings on this relationship. The selected articles were organised into chronological order so to better visualise the evolution of the literature on the subject that was under analysis.

## 3. Results

This section organises and summarises the results obtained in fulfilment of Stage 5 in the development of the scoping review.

### 3.1. Sample Characteristics

The studies included in the review were conducted in 2000–2022. Table A1 shows the sample characteristics (see Appendix A). Most of the study samples were from the United States (*n* = 14; 35.89%), which was followed by Turkey (*n* = 3; 7.69%), China (*n* = 2; 5.12%), Hong Kong (*n* = 2; 5.12%), Canada (*n* = 2; 5.12%), Japan (*n* = 2; 5.12%), Australia (*n* = 2; 5.12%), Taiwan (*n* = 1; 2.56%), Singapore (*n* = 1; 2.56%), South Korea, (*n* = 1; 2.56%), Egypt (*n* = 1; 2.56%), Belgium (*n* = 1; 2.56%), Netherlands (*n* = 1; 2.56%), Iran (*n* = 1; 2.56%), Russia (*n* = 1; 2.56%), Norway (*n* = 1; 2.56%), Croatia (*n* = 1; 2.56%), Spain (*n* = 1; 2.56%) and Israel (*n* = 1; 2.56%). The children ranged in age from two years and two months to six years old. No studies were found from Latin America. Most of the studies included both male and female participants (*n* = 38; 97.43%), and more than half of the studies specify the participants’ ethnicity (*n* = 20; 51.28%). The sample included predominantly White participants. Of the thirty-nine studies, five of them included Asian participants (12.82%), seven of them included Latino participants (17.94%) and seven of them included Black participants (17.94%).

### 3.2. Methodological Differences

Of the 39 international articles included in this scoping review, twenty-five of them had a cross-sectional design (64.10%), seven of them had a longitudinal design (17.94%), and seven of them had a short-term longitudinal design (17.94%). No cross-cultural studies were found.

For the most part, the informants for the data collection were children, parents and teachers (*n* = 29; 74.35%). When the parents were the informants, only one study included couples of the same gender (2.56%), and only one study (2.56%) included a peer nomination procedure. In most of the studies, a composite measure was used to analyse the aggressive behaviour (*n* = 19; 48.71%). Physical aggression (*n* = 21; 53.84%), relational aggression (*n* = 17; 43.58%) and verbal aggression (*n* = 8; 20.51%) were the most frequent forms of aggression. The most frequently analysed one was perpetration (*n* = 36; 94.87%), which was followed by victimisation (*n* = 5; 12.89%) and the combined perpetrator-victim role (*n* = 1; 2.56%). No studies were found which analysed the roles of bystander and defender.

Of the instruments used to measure preschool aggression via the teachers’ reports, the most common ones were: the “Preschool Social Behaviour Scale” [50] (*n* = 8; 20.51%), the “Social Competence and Behaviour Evaluation” scale (SCBE-30) [51] (*n* = 4; 10.25%), the “Child Behaviour Scale” (CBS) [52] (*n* = 4; 10.25%), and the teachers’ reports using items adapted and expanded from Dodge and Coie [53] (*n* = 3; 7.69%). The methods of enquiry also included naturalistic classroom observations (*n* = 4; 10.25%).

There was no overlap in the instruments used in the studies when the parents were the informants of the aggressive behaviour. The instruments were: the parents’ reports using the “Preschool Social Behaviour Scale” [50] (*n* = 1; 2.56%), the “Children’s Social Experiences measure” (CSE) [54] (*n* = 1; 2.56%), the “Children’s Social Behaviour” scale (CSB) [55] (*n* = 1; 2.56%), the “Family and Child Experiences Survey (FACES) interview” [56] (*n* = 1; 2.56%), the “Aggression Scale” developed by Shahim [57] (*n* = 1; 2.56%) and the “Strengths and Difficulties Questionnaire” (SDQ) [58] (*n* = 1; 2.56%).

In terms of the statistical analyses, 97.93% of the articles used both bivariate and multivariate analysis, 2.56% of them used bivariate analyses, 7.69% of them used mediation path analyses, 12.82% of them used moderation analyses and 5.12% of them included interaction analyses. The mediating and moderating variables included: the mothers’ depressive symptoms and the children’s social awareness, authoritative control and perceived parental rejection, child–teacher closeness, the children’s social information process, theory of mind, the aggressive decision-making process, the children’s gender and the children’s effortful control.

### 3.3. Family Variables Included in the Studies

The main family variables included in the studies were those relating to the parenting behaviours, including: psychological control (*n* = 7; 17.94%); parenting styles (*n* = 6; 15.38%); warmth and affection (*n* = 3; 7.69%); physical coercion (*n* = 3; 7.69%); neglect/rejection (*n* = 2; 5.12%); parental self-efficacy (*n* = 2; 5.12%); attachment (*n* = 4; 10.25%); parental values (*n* = 1; 2.56%); social coaching quality (*n* = 1; 2.56%); positive and negative emotion (*n* = 1; 2.56%); emotional regulation (*n* = 1; 2.56%).

Another group of studies looked at the adverse childhood experiences (ACEs; *n* = 7; 17.94%), including: physical abuse (*n* = 5; 12.82%); exposure to violence (*n* = 2; 5.12%); chronic illness within the family (*n* = 1; 2.56%); the death of a family member (*n* = 1; 2.56%); psychopathology (*n* = 6; 15.38%); parent–child conflict (*n* = 3; 7.69%); family stress (*n* = 1; 2.56%); history of maternal smoking during pregnancy (*n* = 1; 2.56%); problems during pregnancy (*n* = 1; 2.56%).

A number of studies also analysed the sociodemographic variables, including: family structure (*n* = 6; 15.38%); education (*n* = 4; 10.25%); occupation (*n* = 3; 7.69%); socioeconomic situation (*n* = 3; 7.69%); unemployment (*n* = 1; 2.56%); the number of siblings (*n* = 3; 7.69%).

Other variables analysed were: the parents’ working hours on weekdays and weekends (*n* = 1; 2.56%), the number of working days (*n* = 1; 2.56%), the length of time spent with the child and the activities they do together (*n* = 1; 2.56%).

There is great variability in the instruments used with the most common ones being: the “Parenting Practices Questionnaire” (PPQ) [59] (*n* = 3; 7.69%) for assessing parental styles, the “Parenting Sense of Competence Scale” (PSOC) [60] (*n* = 2; 5.12%) for measuring parenting self-efficacy, and the “Psychological Control Measure” [61] (*n* = 2; 5.12%) and “Parental Psychological Control” (PPC) [62] (*n* = 2; 5.12%) for measuring psychological control.

### 3.4. Primary Analysis

The results are organised according to the family variables analysed in relation to perpetration and victimisation. Table A1 summarises the main findings of each study on the relationship between preschool aggression and family variables.

#### 3.4.1. Sociodemographic Variables

##### Number of Siblings

Three cross-sectional studies (7.69%) examined the relationship between the number of siblings and preschool aggression. No studies found a positive association between the number of siblings and the perpetration of preschool aggression [63,64,65]. The search did not yield any studies researching the association between the number of siblings and victimisation.

##### Family Structure

Six cross-sectional studies (15.38%) examined the relationship between a single-parent status and preschool aggression. Three of these found no positive association with perpetration [63,64,66]. Among the studies founding positive associations, Honing and Su [67] found differences by gender. In particular, the children whose custodial parent was of the same gender received lower scores on the aggression measure from their teachers than the children whose custodial parent was of a different gender. Jansen et al. [68] found that single parenthood increased the risk of children being perpetrators and perpetrator/victims of preschool aggression. In terms of the type of aggression, Baker et al. [69] found that children in single-parent families have a higher risk of perpetrating proactive relational, reactive relational and reactive physical aggression. In the same vein, Meysamie et al. [65] found an association between single parenthood and relational aggression. Only one study examined the associations between the family structure and victimization [68], and none of them found a positive relationship.

##### Socioeconomic Level

Three cross-sectional and longitudinal studies (7.69%) examined the relationship between the socioeconomic level and aggression in preschool. The studies reveal a positive association between perpetration and low economic status, although there are differences depending on the type of aggression. Jansen et al. [68] reported that socioeconomically disadvantaged children were more at risk of being perpetrators or perpetrator/victims for all forms of aggression. Baker et al. [70] found that children with a low socioeconomic status scored higher for relational aggression but not for physical aggression when they were compared to the children with a high socioeconomic status.

In terms of victimisation, there is no clear picture. Navarro et al. [71] reported that the risk of peer victimisation increases for the children with a low or low-middle socioeconomic status. However, the same relationship was not found by Jansen et al. [68].

##### Level of Educational Attainment

Four cross-sectional studies (10.25%) examined the relationship between parental education attainment and preschool aggression. The results reveal a positive association between perpetration and low parental educational attainment. For example, low parental educational attainment increased the risk of the children being perpetrators or perpetrator/victims [68]. Along the same lines, less educated parents were more likely to have preschool children exhibiting both proactive and reactive aggressive behaviours [66]. With regard to the type of aggression, Meysamie et al. [65] found that a maternal educational attainment below undergraduate level was associated with parent-reported verbal aggression. However, paternal educational attainment was not associated with physical or relational aggression. Amin et al. [72] found lower aggression scores among the children whose mothers were university educated (3.8 vs. 4.8), but these results did not achieve a statistical significance. Only one study looked at the association with victimisation, with Jansen et al. [68] finding that of all of the socioeconomic indicators, only low parental educational attainment was associated with victimisation.

##### Parental Occupation

Three cross-sectional studies (7.69%) examined the relationship between occupation and preschool aggression. Most of the studies show a positive association with perpetration. The children of working mothers show more aggressive behaviour than the children of stay-at-home mothers [65,72]. However, Katsurada [63] does not find the same association. No studies have been found analysing the association between parental occupation and victimisation.

##### Unemployment

Only one cross-sectional study (2.56%) examined the relationship between unemployment and preschool aggression. Family unemployment was associated with an increased likelihood of being a perpetrator or perpetrator/victim in preschool [68]. However, the same association was not found for victimisation.

#### 3.4.2. Adverse Childhood Experiences

As outlined below, most of the studies have examined one single adverse childhood experience in relation to preschool aggression. However, one of the studies that was reviewed (2.56%) examined the cumulative effect of experiencing several adverse childhood experiences. Jiménez et al. [73] examined the longitudinal relationship between preschool aggression and different adverse childhood experiences including child abuse, parental substance use, incarceration and the caregiver being treated violently. Their results showed that an experience of three or more ACEs was associated with an increase in the perpetration of aggressive behaviours.

##### Exposure to Violence

Two short-term longitudinal studies (5.12%) examined the relationship between exposure to violence and preschool aggression. The children that were reported to have witnessed violence within the family were more likely to attribute hostile intentions to their peers, to respond aggressively and to deem socially unacceptable responses appropriate [74]. Along the same lines, an older sibling’s relational and physical aggression was predictive of a younger sibling’s relational and physical aggression towards their peers [11]. No studies have been found which examine the association between exposure to violence and victimisation in preschool.

##### Parent–Child Conflict

Three studies (7.69%), one longitudinal one [75] and two cross-sectional ones [76,77], examined the relationship between parent–child conflict and preschool aggression, yielding mixed results. The longitudinal study showed that children with fewer conflicting family relationships during preschool display less aggressive behaviours in school settings [75]. Ostrov and Bishop [76] found a positive association with relational aggression, but not with physical aggression. However, Farver et al. [77] did not find this association for any types of aggression. For victimisation, no studies examining these relationships have been found.

##### Physical Abuse

Five cross-sectional and longitudinal studies (12.82%) examined the relationship between childhood physical abuse and preschool aggression. All of the longitudinal [74,78,79] and cross-sectional studies [80] found a positive association between physical abuse and perpetration. Some of these studies have found differences depending on the gender of the children and their parents. For example, Ngee Sim and Ping Ong [80] unexpectedly found that paternal spanking was associated with aggression regardless of the child’s gender, while maternal spanking was associated with child aggression only when the child’s perceived rejection value was low. Moreover, maternal spanking was associated with the sons’ aggression, while paternal spanking was associated with the daughters’ aggression only when the rejection value was low. No significant associations have been found between physical abuse and victimisation [71].

##### Parental Criminality

One cross-sectional study (2.56%) examined the relationship between parental criminality and preschool aggression. Having a parent with a criminal record was significantly associated with high levels of child aggression. The association was stronger when was parents were involved in violent and frequent offending [81]. No studies have been found looking at the association between parental criminality and preschool victimisation.

##### Psychopathology

Six studies (15.38%) examined the relationship between mental illness and preschool aggression. In terms of perpetration, the longitudinal studies suggest a positive association [78,81,82], although they find that maternal mental illness plays a greater role than paternal mental illness [81,82]. However, the picture is somewhat less clear for the cross-sectional studies. Jung et al. [83] found that the children of mothers with elevated depressive symptoms were rated by their mothers as being more aggressive than the children of mothers with non-elevated depressive symptoms. However, the teachers reported no significant differences. Farver et al. [77] found no direct association between the mothers’ depressive symptoms and their children’s aggressive behaviour. In relation to victimisation, only one study (2.56%) analysed the association between parental psychopathology and being a victim of aggression in the school environment, but none of them found any significant associations [71].

##### Chronic Disease within the Family

A single cross-sectional study (2.56%) examined the relationship between chronic illness within the family and preschool aggression, reporting that chronic illness within the family is associated with verbal and relational aggression [65]. No studies have been found which looked at the association between chronic illness within the family and victimisation.

##### Death of a Family Member

One cross-sectional study (2.56%) looked at the association between the death of a family member and preschool aggression, finding a positive relationship between the death of a family member and the perpetration of verbal aggression. However, the same association was not found for physical or relational aggression [65]. No studies have been found which examine the association between the death of a family member and victimisation.

##### Problems during Pregnancy

A single longitudinal study (2.56%) examined the association between being a victim of preschool aggression and problems during pregnancy, but none of them found a significant association [71]. No studies were found that looked at the association between perpetration and problems during pregnancy.

##### History of Maternal Smoking during Pregnancy

One cross-sectional study (2.56%) analysed the maternal smoking history during pregnancy in relation to preschool aggression, but it did not find any significant association [65]. No studies were found examining the same relationship for victimisation.

##### Family Stress

Only one of the reviewed studies (2.56%) examined the relationship between family stress and preschool aggression. DeMulder et al. [84] found that higher levels of family stress were only significantly related to anger/aggression in boys. No studies were found examining this same relationship for victimisation.

#### 3.4.3. Parenting Practices

##### Parental Styles

Six studies (15.38%), two longitudinal ones [71,85] and four cross-sectional ones [66,80,86,87], examined the relationship between parenting styles and preschool aggression. With regard to perpetration, Casas et al. [86] found that authoritative and permissive parenting styles were positively related to the children’s relational aggression, and an authoritative parenting style was associated with lower levels of physical aggression in children. Jia et al. [66] found that hostile/coercive parenting was an independent risk factor for both proactive and reactive aggression in the preschool environment. However, other studies have not found a significant direct or indirect association between parenting styles and aggressive behaviour in preschoolers [80,85,87]. In the case of victimisation, only one study analysed this relationship. The results showed that the parental practices characterised by lower norms at the age of three increased the risk of the parents and teachers reporting persistent victimisation at the ages of four and five [71].

##### Physical Coercion

Three studies (7.69%), one longitudinal one [88] and two cross-sectional ones [89,90], examined the relationship between physical coercion in childhood and preschool aggression. All of them found a positive association with perpetration, though there were differences depending on the gender of the children and their parents. For example, Nelson et al. [89] found a positive association between physical coercion and aggressive behaviour, but only in boys. With regard to the parent–child dyad, Lau et al. [88] found differences depending on the parent. Maternal physical coercion was longitudinally associated with both physical and relational aggression. However, paternal physical coercion was longitudinally correlated only with physical, but not relational aggression. As for preschool victimisation, no significant associations with physical aggression have been found [90].

##### Psychological Control

Seven cross-sectional and longitudinal studies (17.94%) examined the relationship between psychological control and preschool aggression. Most of the longitudinal [86,91] and cross-sectional research [61,89] reveals a positive association with the perpetration of both physical and relational aggression. Differences were found depending on the dimensions that were studied within psychological control and the parent–child dyads that were analysed. For example, Nelson et al. [61] found that all of the dimensions of psychological control (shaming/disappointment, constraining verbal expression, withdrawal of love and guilt induction) were associated with physical and relational aggression, with the exception of invalidating feelings, albeit predominantly in same-gender parent–child dyads. However, in a longitudinal study, Lau et al. [92] found that neither maternal nor paternal psychological control was associated with either physical or relational aggression. No studies were found which examined the same relationship with victimisation.

##### Neglect/Rejection

One cross-sectional study (2.56%) examined the relationship between neglect/rejection and preschool aggression. Shim and Kim [90] found that parental neglect/rejection increased the likelihood of peer victimisation in preschool. No studies have analysed the direct relationships between parental rejection and perpetration.

##### Warmth and Affection

Three studies (7.69%), two cross-sectional ones [63,90] and one longitudinal ones [79], examined the relationship between warmth and preschool aggression, with mixed results. Low warmth/responsiveness was the best precursor of later peer aggression in the children with low levels of theory of mind [79]. In the same vein, Katsurada [63] found that children who had more positive physical contact with their parents at home were less likely to be rated as extremely aggressive by their teachers. However, when both of the parents’ physical affection scores were simultaneously inserted into the equation as independent variables, only the father’s score was significant. With regard to victimisation, Shim and Kim [90] found no association between warmth and victimisation in preschool children.

##### Attachment

Four studies (10.25%) examined the relationship between attachment and preschool aggression. In the case of perpetration, the longitudinal [85] and cross-sectional [84,86,93] analyses found a positive relationship between an insecure attachment and aggressive preschoolers. Paschall et al. [85] found that children exposed to detached parenting, which is characterised by the highest level of detachment, a moderate level of negative regard and the lowest level of supportiveness, showed higher levels of aggression at 36 months than the children who were exposed to sensitive parenting. However, there were no significant differences in the aggression levels between the children exposed to detached as opposed to harsh parenting, or between the children exposed to sensitive as opposed to harsh parenting. Casas et al. [86] found that the association between insecure attachment and relational and physical aggression varied with the gender composition of the parent–child dyad. Parent–child closeness moderates the relationship between a lower quality of attachment and aggressive behaviour [93]. No studies have been found which examine this same relationship with victimisation.

##### Parenting Self-Efficacy

Two cross-sectional studies (5.12%) examined the relationship between parental self-efficacy and preschool aggression. These studies found that low perceived parental self-efficacy among mothers was significantly associated with higher levels of aggression and peer victimisation in the school environment [94,95]. This relationship was stronger between parental self-efficacy and peer victimisation [95].

##### Parental Values (Individualism, Collectivism and Verticalism)

A single cross-sectional study (2.56%) involved the relationship between parental values and preschool aggression. The combination of maternal individualistic and collectivistic values was associated with a higher social competence in children, and thus, less aggressive behaviour. This association was not significant for fathers [96]. No studies were found examining this same relationship with victimisation.

##### Social Coaching Qualities (Elaboration, Emotion References and Rule Violation)

A single short-term longitudinal study (2.56%) looked at the social coaching qualities in relation to preschool aggression. After controlling for physical aggression, the children whose mothers used medium or high levels of emotion-focused, elaborative social coaching about relational conflict were less relationally aggressive in the following year. However, the maternal use of rule violation did not explain the individual differences in the children’s use of relational aggression, nor did it moderate the association between time 1 and time 2 relational aggression [97]. No studies examining this relationship have been found for victimisation.

##### Other Emotional Variables

Two studies (5.12%), one cross-sectional one [98] and one longitudinal one [88], examined the relationship between the caregivers’ emotional cues and preschool aggression. Mizokawa and Hamana [98] found a positive association, albeit with differences depending on the gender of the children and their parents. Boys with higher levels of theory of mind and mothers with high negative emotional expression show higher relational aggression in the preschool environment. No effect was found for girls. Contrary to previous studies, Lau and Williams [88] unexpectedly found that higher levels of reappraisal and lower levels of maternal suppression were associated with greater child relational aggression. The authors explained that these unexpected results could be mostly due to cultural factors. They did not find a significant association with paternal emotional regulation. No studies examining these same associations have been found for victimisation.

#### 3.4.4. Other Factors

##### Daily Working Hours and Days of Work

Of the studies that are included in the review, one cross-sectional study examined the association between the fathers’ daily and weekly working hours and preschool aggression [99]. The results indicate that the duration of the fathers’ daily and weekly work and the number of days worked per week were significantly associated with their children’s levels of aggression. The number of hours worked per day was the variable most strongly related to aggressive behaviour. The more hours that they worked, the higher the levels of aggression were among the five and six-year-olds in the school setting. No studies were found analysing this same association for victimisation.

##### Time Spent with the Child

Güngör et al. [99] analysed the association between the time spent with the children and preschool aggression. They found that not spending time with children increases the likelihood of the children behaving aggressively towards their classmates. No studies have been found examining the same association for victimisation.

## 4. Discussion

In recent decades, there has been growing concern to analyse the factors that influence the development of aggressive behaviour in the classroom. The socioecological theory [35] points to explanatory factors in the family environment that could be responsible for the children developing aggressive and victimising behaviours in the school setting. On the basis of this premise, this review aimed to examine the scientific literature in order to answer the following question: *what is known about the family variables related to peer aggression and victimisation in preschool settings?* This scoping review includes 39 peer-reviewed articles from an initial sample of 2002 of them, with the main conclusions of each study having been extracted and summarised.

The review of the studies detailed here indicates that the family variables associated with the perpetration of preschool aggression. Positive associations have been found with: single parenthood [65,68,69]; a low economic level [68,69]; a low parental educational attainment [65,66,68]; unemployment [68]; working mothers [65,72]; exposure to violence [11,74]; parent–child conflict [75,76]; physical abuse [74,78,79,80]; parental criminality [81]; mental illness [78,81,82,83]; chronic disease within the family [65]; the death of a family member [65]; family stress [84]; authoritative and permissive parenting [86]; hostile/coercive parenting [66]; physical coercion [88,89,90]; psychological control [61,86,89,90]; a low level of warmth in children with low levels of theory of mind [79]; insecure attachment [84,85,86,93]; low parental perceived self-efficacy [94,95]; low levels of elaborative, emotion-focused social coaching [97]; high maternal negative emotional expression [98]; higher levels of reappraisal and lower levels of suppression [92]; the duration of the fathers’ daily and weekly working patterns [99]. Negative associations have been found with positive physical contact [63]; custody with same-sex parents [67]; maternal reports of the combination of individualistic and collectivistic values [96].

Overall, the studies reviewed suggest that negative parental behaviours characterised by physical coercion, insecure attachment, low parental perceived self-efficacy and punitive styles are those which are most commonly associated with perpetration. Along the same lines, exposure to traumatic situations and violence affecting the family environment (exposure to violence, physical abuse, parental criminality, chronic illness within the family, the death of a family member, family stress and psychopathology) make a significant contribution to the development of peer aggression at the preschool age. However, there are inconsistent results for other factors such as single parenthood and parental occupation, psychological control, psychopathology and authoritative parenting.

Furthermore, the studies show some sociodemographic factors which increase the risk of perpetration in preschools. The results of this review indicate a positive association between unemployment, low parental educational attainment and economic status with the perpetration of preschool aggression.

Little is known about the family factors and their relationship to victimisation in preschool children. The research has found a significant association with the family variables such as low or low-middle socioeconomic status [71]; low educational attainment [78]; parental practices characterised by lower norms [71]; parental neglect/rejection [90]; low parental perceived self-efficacy [94,95]. These factors may affect the development of mature behaviours such as autonomy, assertiveness and self-confidence [90], as well as competence in conflict resolution with peers and a lack of coping strategies for aggressive behaviour [68,71], which may increase their vulnerability to victimisation. However, it is difficult to draw clear conclusions due to the limited analysis of the family factors in relation to victimisation. Further research is needed to explore the impact of family variables on preschool victimisation.

As they are put in this way, the existing evidence lends support to different theoretical frameworks, such as Bowlby’s attachment theory [100], the social learning theory and the intergenerational transmission of violence [101], and the family-relational schema [102]. These theoretical frameworks conclude that children develop behaviours and internal cognitive models of both the self and relationships with other people through their primary caregivers. Positive relationships during childhood with adults who provide affection and a secure base for exploring the social environment are fundamental to positive socioemotional development. However, the children with poor quality family relationships raised in socially disadvantaged environments may be at risk of learning negative relationship patterns, thus leading to aggressive behaviour or to them becoming victims of peer aggression [103].

The results of the studies reviewed also underpin the importance of considering the differential susceptibility of children to parenting behaviours. In particular, the evidence suggests that the factors underlying perpetration and victimisation in preschool aggression vary depending on the child’s gender and the parent–child dyads [61,80]. However, not all of the studies analysed the gender differences or include the data on parent–child dyads, making it difficult to draw clear conclusions. In line with Narayana and Naerde [82], future research should shed light on the mechanisms underlying the transmission of this risk from parents to children and examine how this may differ between the mothers and fathers at different developmental stages.

While it was not specifically an objective of this review, as noted in the introduction, another critical question for the researchers and practitioners has been whether it is possible to extend the concept of bullying to the study of aggressive behaviours occurring during early childhood [104,105,106]. Only four (10.25%) of the studies reviewed have used the term “bullying” to refer to aggressive behaviour in preschool [63,68,71,91]. However, only two of them included specific measures of bullying in the description of their method, or they have analysed the repeated nature of such behaviours [68,91]. Nevertheless, both of the studies measured bullying only through the teachers’ reports, and neither of them analysed other peripheral roles such as defenders or included measures to analyse the intentionality, stability and power inequality characterising these behaviours [20]. Nor did they include a definition of bullying to clarify these dimensions for the teacher respondents. It is therefore difficult to arrive at firm conclusions on the relevance of discussing bullying at the preschool stage, and more research is needed. The qualitative research has shown that children, as well as teachers and parents, identify bullying behaviour during childhood. González-Moreno et al. [19], for example, found that from an early age, children are able to identify bullying behaviours and attribute meanings to these behaviours in very similar manners to those of adolescents. However, these researchers note, nonetheless, that such behaviours could be termed ‘proto-bullying’ and identifying them could be key to preventing bullying behaviour at later stages.

### Limitations and Recommendations

Despite the merits of this review, some important limitations do exist, first and foremost, there are those that are related to the search. On the one hand, only articles in English and Spanish were included, thus potentially excluding relevant studies that were published in other languages. Moreover, the use of a limited number of databases and the search terms themselves may have led to the exclusion of some studies addressing the research question posed.

Secondly, the diversity of the studies included in this review means that constructs such as aggressive behaviour itself, parenting styles or certain parenting practices may take on varying meanings in different cultural and social contexts. This makes the comparison difficult, and it means that the results should be treated with caution, taking into account these potential differences. This being the case, it is worth noting that in those studies where the information from parents and teachers was available in relation to the children’s aggressive behaviour, there were differences in the levels of reported aggression. Although engaging a range of informants is important when one is analysing aggressive behaviour, the parents’ and teachers’ reports may be affected by different factors. These include the fact that children behave differently in different environments and with different people. Moreover, aggressive behaviours, such as relational behaviours, may be easier to perceive in a context of ongoing peer relationships, such as in school, as opposed to other contexts outside the home environment, such as parks, playgrounds and so on. Alternatively, the informants may show lower or higher tolerance for aggressive behaviours [83]. We believe that the teachers’ reports may be more descriptive of the aggression in the school environment because of the time they share with their pupils and their knowledge of the relationship dynamics among them. For this reason, greater consideration has been given to the teachers’ reports of aggressive behaviour in order to draw conclusions about the relationship between the family variables and preschool aggression. This is presumably a limitation of this review.

Thirdly, and again, when one is making comparisons and drawing conclusions, it should be borne in mind that the measurement instruments that are used differ from one study to the next. In this regard, the family factors were measured mainly through self-reports by the fathers, mothers or by both of the parents. Self-reports have the advantage of capturing an overall picture of parenting, but they may also be vulnerable to social desirability and recall biases [91]. The literature suggests that supplementing these measures with direct evaluations of parenting is likely to increase the predictive power of these constructs [87]. Alternatively, as suggested by Nelson et al. [61], it would be appropriate to supplement them with the information provided by the children about their parents’ behaviour, and whenever possible, to find out the spouse/partner’s view. The fact is that virtually no studies have been found with child-reported information raises the likelihood of the researchers overlooking some cases of victimisation and perpetration. Finally, the tools used to measure the aggression and family variables differed between the articles, which also limits comparability between the studies and the opportunity to draw more accurate conclusions.

Fourthly, the articles in this review are mostly cross-sectional in design, so any conclusions about the influence of the family factors on aggressive behaviour must acknowledge the difficulties in establishing the causality of the factors. It would be advisable to conduct longitudinal studies over extended periods of time and to plan studies with large sample sizes. This would help in gaining a better understanding of the association between the family factors and preschool aggression. In addition, most of the samples involved middle-class, White participants. The inclusion of comparison groups of families from different socioeconomic backgrounds and cross-cultural studies would permit a more direct examination of the specific effects of parenting culture on the children’s psychosocial development. In this sense, the inconsistencies in certain findings may be partly attributable to the limitations of the analytical approaches and the sample that were used. For example, there has been little control for the other individual, family and contextual factors that mediate or moderate the relationship between the perpetration or victimisation behaviours and the family variables. The future research should try to identify such factors that may play a role in mediating or moderating this relationship.

Nevertheless, in answer to the question that is posed in this review, it is clear that there are family variables related to peer aggression and victimisation in the preschool environment. However, most of the current intervention strategies for the prevention of school-based perpetration of aggression and victimisation neglect the role of family relationships [107]. The findings of the studies contained in this review suggest that intervention programmes aimed at reducing the children’s personal risk factors should be combined with counselling and training programmes for the parents. In general, training programmes can help the parents to develop healthier relationship patterns with their children, with the goal of making the parents aware that their relationship with their child is the basis for the child’s psychosocial development [87]. The programmes should aim to promote positive parenting practices, foster the children’s perceived support and affection, and establish patterns promoting intrafamilial communication and closeness [108]. Not only should such interventions aim to promote positive parenting skills, but they should also help the parents to deal with problems in their personal lives, including medical care, stress, job training, developing communication skills and identifying available community resources [109], and seeking support from teachers [110].

Furthermore, the existing research suggests that the outcomes of aggressive behaviour prevention programmes in the school environment could be improved by focusing on the at-risk groups identified in preschools, and not only by targeting primary and secondary schools [71]. In line with Paschall et al. [85], we further recommend that intervention programmes incorporate specific additional support for the parents who face sociodemographic risk factors linked to adverse social and economic conditions, and who are exposed to other harmful events such as domestic abuse, criminality, psychopathology and so on. The types of preventive interventions that appear to be most effective for the at-risk groups are those that start early and are holistic, involving a systemic approach with multiple interventions that touch on the individual, family and community levels [109].

## 5. Conclusions

This scoping review shows that different family environmental factors increase the risk of developing perpetration and victimisation behaviours in the preschool setting. However, the relationship between the family variables and aggression in preschool children is complex, and it may be mediated by other co-existing factors, such as gender, teacher closeness and the parent–child dyads, or personal factors such as theory of mind and the children’s effortful control. The results have important implications for prevention and intervention programmes to prevent aggression from spilling over into later educational stages.

## Figures and Tables

**Figure 1 ijerph-19-15556-f001:**
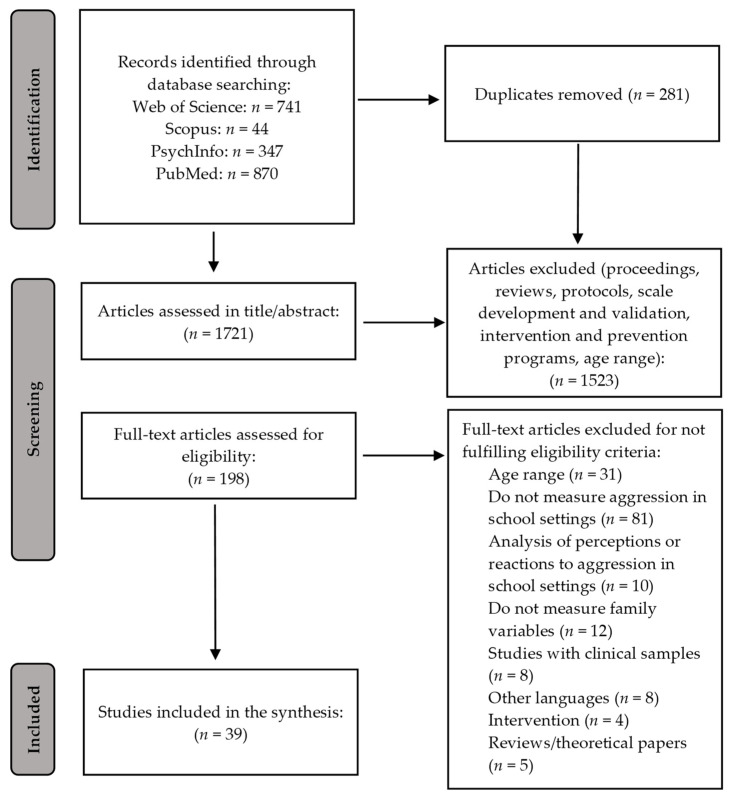
Selection procedure flow chart.

**Table 1 ijerph-19-15556-t001:** Search terms used.

Search Terms
1. (“preschool *” OR “preschool children” OR “preschool-aged children” OR “early child*” OR “2–6 yrs”).ti
2. (“victim *” OR “perpetrat *” OR “aggress *” OR “bullying” OR “bully” OR “bully/victim” OR “peer aggress *” OR “peer violen *” OR “peer victim *” OR “peer abus *”).ti
3. (“family” OR “families” OR “family context” OR “parent *” OR “mother” OR “father”).ti
4. Searches 1, 2 and 3 were performed in each database.

**Table 2 ijerph-19-15556-t002:** Summary of the inclusion and exclusion criteria.

Inclusion	Exclusion
Qualitative and quantitative research published in peer-reviewed journals.	Articles describing interventions or prevention programmes, literature reviews, systematic reviews, conference papers, doctoral theses and newspaper articles.
Participants aged two to six years old.	Participants who were less than two or more than six years old.
Participants in the study had to belong to general populations.	Clinical samples or subgroups.
Research analysing aggressive peer interactions in school environments.	Research analysing aggressive interactions beyond peer relationships in school settings.
Research examining the relationship of preschool aggression using family or parental variables as predictor variables.	Research examining the relationship of preschool aggression not using family or parental variables as predictor variables.
Published in English and Spanish.	Published in languages other than English and Spanish.

## Data Availability

All data generated as part of this study are included in the article.

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
