# Peer review of "Families, Parenting and Aggressive Preschoolers: A Scoping Review of Studies Examining Family Variables Related to Preschool Aggression"

_ijerph, 2022, doi:10.3390/ijerph192315556_

Round 1

Reviewer 1 Report

REVIEW

Families, Parenting and Aggressive Preschoolers: A Scoping Review of Studies Examining Family Variables Related to Preschool Aggression.

This is a very interesting manuscript that should be published, considering: the relevance and pertinence of the topic and the high quality of the scoping review which had followed with utmost rigor the PRISMA guidelines.

Some Comments and Minor Suggestions

Results

·        Sample Characteristics (p.6) – As regards the geographical origin of the samples of the reviewed studies, the authors present the number of studies in each country in descending order. I believe there is a mistake in the text, as Turkey (with three studies) is out of the descending order (line 220).

·        Family Structure (p.8) – The authors refer that, in a total of six studies, four found no positive association between single-parent 301 status and preschool aggression (line 302) which suggest that positive associations were reported in two of the six studies. However, in the following lines (303-311), it is reported that associations were found in three studies (69, 65 and 66). I suggest that the authors clarify this issue.

·        Physical Abuse (p.9) – Just a comment: I was really surprised with the followed results of Ngee Sim and Ping Ong study (384-387): “maternal spanking was associated with child aggression only when the child's perceived rejection is low”; “while paternal spanking was associated with daughters' aggression only when rejection was low”. Trying to understand those results, I read Sim and Ong paper, but I think the authors explanations are not so clear: “Although we had predicted that greater rejection would be associated with a stronger punishment-aggression link, we found positive relations for mother caning and father slapping at low rejection, but no relations at moderate or high rejection. This pattern of results suggests that when parents are perceived as less rejecting, learning or modeling processes (e.g., Bandura et al., 1963) may actually come into play when parents use physically punitive acts (cf. Gunnoe & Mariner, 1997). This is a possibility that future studies can explore” (Sim & Ong, 2005, p.95).

I suggest that the authors of the present manuscript point those results as unexpected even to Sim and Ong.

·        Attachment (p.12; line 493) - The end point is missing after “(…) exposed sensitive parenting”

·        Other Emotional Variables (p.12) – The described results (“Lau and Williams [88] found that higher levels of reappraisal and lower levels of maternal suppression were associated with greater child relational aggression” – lines 528-530) are contrary to several other studies, which is, according to Lau and William, mostly due to cultural factors. As so, I suggest that the authors of the present manuscript point those results as unexpected considering previous studies (and partially unexpected even to Lau and William) and add a brief explanation on it.

Discussion

·        (p.13; lines 558-576) - The family variables associated with the perpetration of preschool aggression could be organized according to the direction of the associations (positive vs. negative) and some associations should be clearer (paternal/maternal custody; individualistic and collectivistic values).

·        (p.13; lines 576-577) – “Furthermore, studies show that sociodemographic factors increase the risk of perpetration in preschool” should be replaced by: ““Furthermore, studies show some sociodemographic factors that increase the risk of perpetration in preschool”.

·        (p.13; lines 580-581) – It should be paragraph before “However, there are inconsistent results for other factors such as single-parenthood and parental occupation, psychological control, psychopathology and authoritative parenting.” In the continuity of the previous sentence, without a paragraph, it seems that the authors are still referring to sociodemographic factors.

Author Response

First, we are extremely grateful for the comments from the reviewers, which we believe have contributed to significant improvements in the paper.

We have commented on each piece of feedback separately and we have attached an updated version of our paper after revisions.

Changes introduced in the manuscript haven been tracked.

Reviewer 1

Comment 1: Sample Characteristics (p.6) – As regards the geographical origin of the samples of the reviewed studies, the authors present the number of studies in each country in descending order. I believe there is a mistake in the text, as Turkey (with three studies) is out of the descending order (line 220).

      Authors’ response: you are right. Thank you. We have relocated Turkey number of studies.

Comment 2: Family Structure (p.8) – The authors refer that, in a total of six studies, four found no positive association between single-parent 301 status and preschool aggression (line 302) which suggest that positive associations were reported in two of the six studies. However, in the following lines (303-311), it is reported that associations were found in three studies (69, 65 and 66). I suggest that the authors clarify this issue.

      Authors’ response: thank you for pointing this out. Three studies found a positive association and three failed to find the same association with perpetration. As we added the study not founding and association with victimization (a study that examined both perpetration and victimization) the reading was awkward. We have rewritten the paragraph adding the study analyzing victimization at the end.

Comment 3: Physical Abuse (p.9) – Just a comment: I was really surprised with the followed results of Ngee Sim and Ping Ong study (384-387): “maternal spanking was associated with child aggression only when the child's perceived rejection is low”; “while paternal spanking was associated with daughters' aggression only when rejection was low”. Trying to understand those results, I read Sim and Ong paper, but I think the authors explanations are not so clear: “Although we had predicted that greater rejection would be associated with a stronger punishment-aggression link, we found positive relations for mother caning and father slapping at low rejection, but no relations at moderate or high rejection. This pattern of results suggests that when parents are perceived as less rejecting, learning or modeling processes (e.g., Bandura et al., 1963) may actually come into play when parents use physically punitive acts (cf. Gunnoe & Mariner, 1997). This is a possibility that future studies can explore” (Sim & Ong, 2005, p.95).

I suggest that the authors of the present manuscript point those results as unexpected even to Sim and Ong.

Authors’ response: we appreciate your recommendation. We have pointed that resulted were unexpected.

Comment 4: Attachment (p.12; line 493) - The end point is missing after “(…) exposed sensitive parenting”

Authors’ response: thank you for detailed revision. We have corrected the typo.

Comment 5: Other Emotional Variables (p.12) – The described results (“Lau and Williams [88] found that higher levels of reappraisal and lower levels of maternal suppression were associated with greater child relational aggression” – lines 528-530) are contrary to several other studies, which is, according to Lau and William, mostly due to cultural factors. As so, I suggest that the authors of the present manuscript point those results as unexpected considering previous studies (and partially unexpected even to Lau and William) and add a brief explanation on it.

Authors’ response: Thank you for pointing this out. We have rewritten the sentence explained that the results found by Lau and William were contrary to previous research and unexpected. These contradictory results were explained

Comment 6: Discussion (p.13; lines 558-576) - The family variables associated with the perpetration of preschool aggression could be organized according to the direction of the associations (positive vs. negative) and some associations should be clearer (paternal/maternal custody; individualistic and collectivistic values).

Authors’ answer: thank you. This is a good idea. We have now indicated positive and negative associations. Associations with parents’ custody and values have been clarified.

Comment 7 (p.13; lines 576-577) – “Furthermore, studies show that sociodemographic factors increase the risk of perpetration in preschool” should be replaced by: ““Furthermore, studies show some sociodemographic factors that increase the risk of perpetration in preschool”.

      Authors’ answer: thank you for this. We have changed the sentence following your advice.

Comment 8: (p.13; lines 580-581) – It should be paragraph before “However, there are inconsistent results for other factors such as single-parenthood and parental occupation, psychological control, psychopathology and authoritative parenting.” In the continuity of the previous sentence, without a paragraph, it seems that the authors are still referring to sociodemographic factors.

Authors’ answer: you are right. We have moved up that sentence and paragraph the sentence about sociodemographic factors.

Reviewer 2 Report

Thank you for the opportunity to review this manuscript.

The paper reports a scoping review of studies examining family variables related to preschool aggression.

The title is appropriate and indicates the main issue of the paper. The abstract provides a complete summary of the content of the manuscript. At line 21 the word “Method” should be in italics. The keywords of the abstract are adequate. The reasons of the scoping review are well-explained. The review employs concepts previously used in related literature.

I find that the introduction could be improved. It provides sufficient background information for readers not in the specific field to understand the objectives of the review. However, I did not read anything about the literature that already addressed the main issue of the review, i.e. the family variables related to preschool aggression. At lines 99-100 the authors say that the family context is fundamental for children’s development. I would add here few lines introducing the main family variables related to preschool aggression.

The objectives of the review are clearly and explicitly defined both in the abstract and in the main text. The methods used are rigorous and appropriate to the aims of the study. Sufficient information is provided for a capable researcher to reproduce the review because methods and instruments are presented clearly.

The results are clearly presented in an appropriate format, in particular the table A1 in appendix 1 shows essential data that could not be easily summarized in the text. These data are also easy to read and interpret. The number and scientific quality of the references cited are adequate.

The discussion of the results is well-organized and the authors seem aware of the limitations of their review. Sufficient space was given to the hands-on practical implications of the review.

In conclusion, this is a well-written paper and the study presented in it has a good scientific quality. The topic of the review is timely and would be of interest to the readership of the Special Issue "Children’s Health: Feature Review Papers" of IJERPH. However, I would change a little the introduction as I suggested before.

Author Response

First, we are extremely grateful for the comments from the reviewers, which we believe have contributed to significant improvements in the paper.

We have commented on each piece of feedback separately and we have attached an updated version of our paper after revisions.

Changes introduced in the manuscript haven been tracked.

Reviewer 2

Comment: I find that the introduction could be improved. It provides sufficient background information for readers not in the specific field to understand the objectives of the review. However, I did not read anything about the literature that already addressed the main issue of the review, i.e. the family variables related to preschool aggression. At lines 99-100 the authors say that the family context is fundamental for children’s development. I would add here few lines introducing the main family variables related to preschool aggression.

Authors’ response: thank you very much for your kind review. Following your advice we have added a few lines introducing the main family variables related to preschool aggression.